# Implications of Salinity and Acidic Environments on Fitness and Oxidative Stress Parameters in Early Developing Seahorses *Hippocampus reidi*

**DOI:** 10.3390/ani12223227

**Published:** 2022-11-21

**Authors:** Mario D. D. Carneiro, Sergio García-Mesa, Luis A. Sampaio, Miquel Planas

**Affiliations:** 1Department of Ecology and Marine Resources, Institute of Marine Research (CSIC), 36208 Vigo, Spain; 2Laboratório de Piscicultura Estuarina e Marinha, Instituto de Oceanografia, Universidade Federal do Rio Grande–FURG, Rio Grande 96210-030, Brazil; 3Department of Zoology, University of Granada, Campus Universitario de Fuentenueva, 18071 Granada, Spain

**Keywords:** *Hippocampus*, seahorse, acidification, salinity, oxidative stress, RAS

## Abstract

**Simple Summary:**

The main aim of the present study was to assess the effects of acidification (pH 6.5 vs. pH 8.0) under two salinity conditions (brackish water—BW vs. seawater—SW) on the development and fitness (oxidative stress) of early developing seahorses (*Hippocampus reidi)*. The growth of juveniles reared in BW was impaired at pH 6.5, and the levels of superoxide dismutase and DT-diaphorase, as well as the oxidative stress index, increased compared to SW juveniles. However, survival and growth at pH 6.5 decreased in the former. These results suggest higher overall performance and optimal fitness in juveniles reared in seawater under acidic conditions (pH = 6.5).

**Abstract:**

Water acidification affects aquatic species, both in natural environmental conditions and in ex situ rearing production systems. The chronic effects of acidic conditions (pH 6.5 vs. pH 8.0) in seahorses (*Hippocampus* spp.) are not well known, especially when coupled with salinity interaction. This study investigated the implications of pH on the growth and oxidative stress in the seahorse *Hippocampus reidi* (Ginsburg, 1933), one of the most important seahorse species in the ornamental trade. Two trials were carried out in juveniles (0–21 and 21–50 DAR—days after the male’s pouch release) reared under acid (6.5) and control (8.0) pH, both in brackish water (BW—salinity 11) and seawater (SW—salinity 33). In the first trial (0–21 DAR), there was no effect of pH on the growth of seahorses reared in SW, but the survival rate was higher for juveniles raised in SW at pH 6.5. However, the growth and survival of juveniles reared in BW were impaired at pH 6.5. Compared to SW conditions, the levels of superoxide dismutase and DT-diaphorase, as well as the oxidative stress index, increased for juveniles reared in BW. In the second trial, seahorse juveniles were reared in SW at pH 8.0, and subsequently kept for four weeks (from 21 to 50 DAR) at pH 6.5 and 8.0. The final survival rates and condition index were similar in both treatments. However, the growth under acidic conditions was higher than at pH 8.0. In conclusion, this study highlights that survival, growth, and oxidative status condition was enhanced in seahorse juveniles reared in SW under acidic conditions (pH = 6.5). The concurrent conditions of acidic pH (6.5) and BW should be avoided due to harmful effects on the fitness and development of seahorse juveniles.

## 1. Introduction

Fishes can be exposed to significant salinity and pH alterations in nature as well as along the production cycle in captivity when the rearing facilities are located in unstable coastal areas, or as a consequence of some management practices (i.e., prophylaxis, disease treatments, or transportation) [1,2,3,4]. The impact of environmental acidification may be detrimental, especially in early developmental stages (i.e., embryos, larvae, and early juveniles) [5,6,7,8].

Many marine ornamentals are produced far from coastal areas in order to avoid high land costs and to improve biosecurity and commercialization [9,10]. Hence, the use of recirculating aquaculture system (RAS) could potentially be advantageous. The use of RAS is continuously increasing, disseminating aquaculture facilities everywhere [11]. However, the environmental conditions in these systems may fluctuate considerably, especially pH, which may reach undesirable levels. Salinity and pH are pivotal factors in fish cultivation due to their implications in the physiological condition of fishes [12,13,14,15,16]. Water acidification in RAS is a consequence of alkalinity reduction and H+ released by the nitrification process [17]. Compared to biofilters in seawater, nitrification performance in freshwater systems is at least 60% higher [11].

The production in captivity of vulnerable or endangered species, such as seahorses, is an alternative method to reduce the exploitation of wild populations, ensuring the traceability of the product in the market [18]. Currently, most seahorses legally traded as ornamental specimens originate from captive breeding [19]. However, only a few species are commercially raised in captivity for ornamental aquaria [20].

Wild populations of seahorses (Genus *Hippocampus*; Family Syngnathidae) are threatened due to intense captures for Traditional Chinese Medicine (dried specimens) and the aquarium trade [21]. The average annual volume of seahorse trade reported in the Convention on International Trade in Endangered Species of Wild Fauna and Flora (CITES) from 2004 to 2011 was estimated at 5,7 million dried specimens and 116,000 live individuals [19]. However, 37 million dried seahorses from bycatch are traded annually [22]. In light of these facts, all seahorse species were considered vulnerable by the IWT (Illegal Wildlife Trade), being included in Appendix II of CITES since 2002 [23].

Seahorses can be raised in RAS [24,25], including multi-trophic rearing systems, along with shrimps and oysters [26]. The tropical species *Hippocampus reidi* (Ginsburg, 1933) is one of the most important seahorses in the ornamental trade [27,28]. The isosmotic point of the species is reached at 11.67 salinity. Under brackish water conditions (10–20 salinity), the growth is enhanced compared to that when rearing in seawater (30–35 salinity) [29]. Growth and survival enhancement in brackish water has also been reported in other seahorse species, such as *H. abdominalis* [30]. However, other seahorse species did not respond similarly. For example, *H. erectus* displayed higher sensibility to ammonia as well as downregulation in oxidative enzymes when exposed to low salinities [31].

In the temperate seahorse *H. guttulatus*, an increase in temperature resulted in increased oxygen consumption at both pH 8.0 and 7.5 (normocapnia vs. hypercapnia conditions) [32] as a result of the mobilization or use of energy sources for fitness maintenance [33,34,35]. Therefore, changes in salinity and pH may also alter energy metabolism, generating reactive oxygen species (ROS) [36,37,38].

Seahorses display high antioxidant activities; however, the effects of varying culture conditions on their antioxidant state are poorly known, and deserve special assessment [39]. Changes in the biochemical status of fishes might be a consequence of environmental alterations, which could promote oxidative stress (i.e., an imbalance between prooxidants and antioxidants molecules) and damages in cellular macromolecules (e.g., lipids, proteins, and DNA) or pathway changes [36,40,41]. In *H. reidi*, the oxidative status (total phenolic content, metal chelating activity, DPPH radical scavenging activity, and ferric reducing antioxidant power) were significantly higher at 15 and 20 salinities [39]. However, a previous investigation revealed that acute exposure to an acidic pH of 5 in brackish water (salinity 11) was stressful to juveniles [42]. However, chronic responses and potential adaptive responses to acidic pH and salinity alterations are still not well understood.

The present study aimed to assess the biological and physiological age-dependent responses of *H. reidi* juveniles exposed to acidic environments under two salinity conditions: brackish water and seawater. Understanding the impact of these conditions on the oxidative status in developing juveniles will contribute to improving the rearing conditions for this species, especially when raised in RAS.

## 2. Materials and Methods

### 2.1. Bioethics

Animal maintenance and manipulation practices were conducted in compliance with all bioethics standards on animal experimentation of the Spanish government (Real Decreto 1201/2005, 10 October 2005) and the Regional Government Xunta de Galicia (REGA ES360570202001/16/EDU-FOR07/MPO01).

### 2.2. Live Prey Culture

Copepods and *Artemia* (nauplii, enriched metanauplii, and enriched adults) were used to feed seahorses. Copepods *Acartia tonsa* were cultivated in 700 L tanks at 26–27 °C in seawater, at initial densities of 1 copepod/mL^−1^. They were fed every two days on the microalgae *Rhodomonas lens* (10^3^ cells mL^−1^). Siphoning of the culture tanks and water renewals (10% of the total volume) were carried out three times per week.

*Artemia* cysts (MC450; Iberfrost, Spain) were incubated at 28 °C for 20 h in 20 L units. Newly hatched nauplii were collected on a 125 μm mesh, then gently rinsed with tap water and transferred to 20 L units for metanauplii production (100 *Artemia* mL^−1^). Metanauplii of several ages (1–4 days) and sizes were enriched twice daily on a mixture consisting of live microalgae *Phaeodactylum tricornutum* (10^7^ cells mL^−1^), red pepper (0.015 g L^−1^; Bernaqua, Belgium), and dried spirulina (0.03 g L^−1^; Iberfrost, Spain) [43,44]. Adult enriched *Artemia* was grown in 100 L units at 26–28 °C with aeration and constant lightning. A long-time enrichment (3–6 days) was carried out with adult *Artemia* from day 16 onwards, on a mixture consisting of live microalgae *P. tricornutum* and *Isochrysis galbana* (10^7^ cells mL^−1^), red pepper (0.015 g L^−1^; Bernaqua, Belgium), and dried spirulina (0.03 g L^−1^; Iberfrost, Spain) [43].

### 2.3. Seahorse Breeding

Adult seahorses *H. reidi* were maintained in ad hoc aquaria [45] at Instituto de Investigaciones Marinas (IIM-CSIC) in Vigo (Spain). Three aquaria sub-units of 160 L each (85 cm height × 75 cm length × 50 cm width) working in a RAS were used as husbandry and breeding aquaria. The aquaria were supplied with filtered (5 µm) and UV-treated seawater. A partial daily water exchange (10–15% of total volume) was applied. Broodstock seahorses were maintained at 33 ± 1 salinity, pH 8.1 ± 0.1, 26 ± 1 °C temperature, and a 14L:10D photoperiod [25]. The seahorses were fed twice daily on live long-time enriched adult *Artemia* and frozen Mysidacea *Neomysis sp.* (Ocean Nutrition, Spain).

### 2.4. Experiments

#### 2.4.1. Trial 1: Effect of pH and Salinity on Seahorse Juveniles

Newborn *H. reidi* were exposed to different pH levels (6.5 and 8.0) in brackish water (BW; 11 salinity) and seawater (SW; 33 salinity). Since a unique brood could not provide the necessary number of seahorses for the whole study, the trial was carried out using two broods (one brood for each salinity level). Newborns were transferred directly from the breeding aquaria to six pseudo-Kreisel aquaria (30 L), each one stocked with 120 fish [46]. The water conditions during the experiment are provided in Table 1, and the photoperiod regime was set to 12L:12D. Three pseudo-Kreisel aquaria were maintained at pH 8.0, whereas the other three were adjusted and subsequently maintained at pH 6.5. For the maintenance of pH 6.5, water was prepared at pH 6.5 and strongly aerated for 24 h before water renewal. The whole RAS unit was filled with water adjusted at the desired pH level for at least 24 h before the onset of the experiment. This procedure guaranteed the elimination of excess CO_2_, thus assuring that the results of the experiment reflected the desired pH instead the toxic effect of CO_2_ [47]. Each group of three aquaria worked as a RAS system, including a sump with pump, heaters, chiller, UV filtration, and biological filters (perforated plastic bio-balls) [44,46].

The pH levels were regularly monitored and adjusted by adding pre-tested volumes of HCl solutions (3% HCl, 0.36 M) to the water used for daily water renewal. Water conditions were monitored twice daily, including pH (Crison, Micro pHmeter 2001, Barcelona, Spain), salinity (Atago S/Milli-E, Tokyo, Japan), dissolved oxygen, temperature (Hach, HQ40d, Loveland, Colorado, CO, USA), and alkalinity [48]. Total ammoniacal nitrogen (TAN = NH_4_^+^ + NH_3_^−^) was measured by spectrophotometry (Cecil, spectrophotometer CE 3040, Cambridge, UK) [49]. Nitrite (NO_2_^−^) and nitrate (NO_3_^−^) were analyzed by a segmented flow analyzer (, Futura, Italy) [50]. All nitrogen compounds were checked twice a week (Table 1).

This trial lasted for 21 days, the age at which the juveniles undergo important morphological and histological changes [51], as well as epigenetic processes involved in the transition from a planktonic to a benthonic lifestyle [52]. The juveniles were fed twice daily (1–3 prey mL^−1^) on copepods *A. tonsa,* retained by 125 µm mesh size 1 to 5 days after the male’s pouch release (DAR); a mixture of copepods (*A. tonsa*), filtered by 180 µm mesh and *Artemia* nauplii (6–10 DAR); or *Artemia* nauplii and metanauplii enriched for 24 h (1:1) (11–21 DAR). At days 0, 2, 7, 14, and 21 DAR, seahorse juveniles were sampled for analytical and biochemical procedures as described below.

Dead seahorses were removed daily (8:00 am and 3:00 pm) from the aquaria and counted. Wastes and uneaten food were removed by siphoning the bottom of the aquaria.

#### 2.4.2. Trial 2: Effect of Acidification on Seahorse Juveniles Reared in SW at pH 8.0

This trial was carried out with 21 DAR juveniles, produced as indicated in trial 1 (SW and pH 8.0). A total of 204 juveniles were transferred to 6 pseudo-Kreisel aquaria (34 juveniles in each aquarium; 1.1 juveniles L*^−^*^1^) filled with SW at pH 6.5 (acidic environment) or pH 8.0 (control). Three aquaria were used for each pH level. Seahorses were reared for four weeks and fed on *Artemia* metanauplii enriched for 48, 72, or 96 h, depending on the size of the seahorses. The feeding schedule was as follows: 48-72 h enriched *Artemia* metanauplii for 21–30 DAR, 72–96 h enriched metanauplii for 31–45 DAR, and 96 h enriched metanauplii from45 DAR onwards. The water conditions were monitored and adjusted as described for trial 1 (see Table 2). The juveniles were measured and weighed at the start (21 DAR), middle (two weeks; 35 DAR), and end (four weeks; 49 DAR) of the trial.

### 2.5. Biochemical Analyses

Biochemical analyses were conducted in fish from trial 1. Seven enzymatic activities were assayed: superoxide dismutase (SOD), DT-diaphorase (DTD), catalase (CAT), glucose 6-P dehydrogenase (G6PDH), glutathione peroxidase (GPx), glutathione reductase (GR), and glutathione-S transferase (GST).

At 0, 2, 7, 14, and 21 DAR, samples of 40, 15, 15, 10, and 10 juveniles were taken, respectively. The juveniles were euthanized in a MS-222 immersion (100 mg L^−1^), flash-frozen in liquid nitrogen, and stored frozen (−80 °C). Each sample was divided into two sub-samples. A sub-sample comprising approximately 70% of the total biomass was homogenized (1:9—*w*:*v*) (Poly Tron, PT 2100) in ice-cold 100 mM Tris-HCl buffer containing 0.1 mM EDTA and 0.1% (*v*:*v*) Triton X-100, pH 7.8. After centrifugation (30,000 g for 30 min at 4 °C) (SIGMA, 3K30), the supernatant was stored at −80 °C for further protein, enzymes, TBARS, and TEAC analyses. The other sub-sample (30%) was homogenized in 10 mM HCl and 1.3% SSA buffer (1:9—*w*:*v*). After centrifugation (20,000 g for 10 min at 4 °C), the supernatant was stored at −80 °C for further glutathione content (GSSG and GSH) determination.

All analyses were performed in duplicate at 25 ± 0.5 °C in 96-well microplates (UVStar, Greiner Bio-One, Frickenhausen, Germany), in a microplate reader (Bio-Tek PowerWave, Santa Clara, CA, USA). The optimal substrate concentration to measure the maximal specific activity for each enzyme was established in preliminary assays. The enzymatic reaction was initiated by the addition of homogenate.

Except for SOD, one unit or milliunit of enzymatic activity was defined as the amount of enzyme required to transform 1 µmol or nmol of substrate min^−1^ under the conditions defined for each assay. Soluble protein content was determined using bovine serum albumin as standard [53], and used to estimate enzyme-specific activity.

Superoxide dismutase (SOD) was assessed by the ferricytochrome C method using xanthine/xanthine oxidase as the source of superoxide radicals [54]. DT-diaphorase (DTD) activity was measured in the reaction mixture containing DCPIP (2,6-dichlorophenol indophenol) and NADH (Nicotinamide adenine dinucleotide). The control reaction contained distilled water instead of sample extract. The DTD activity was determined as the difference between sample and control readings at 600 nm [55].

Catalase (CAT) activity was achieved by measuring the decrease in H2O2 concentration in a reaction mixture containing 50 mM potassium phosphate buffer (pH 7.0) and 10.6 mM H2O2 (freshly prepared) at 240 nm [56].

Glucose 6-P dehydrogenase (G6PDH) was carried out with some modifications. The change in absorbance of NADPH (Nicotinamide adenine dinucleotide phosphate) at 340 nm was monitored in order to determine G6PDH, NADP, and glucose-6-phosphate [57,58].

Glutathione peroxidase (GPx) was indirectly measured using cumene hydroperoxide as substrate and spectrophotometrically monitoring NADPH consumption at 340 nm [59].

Glutathione reductase (GR) was assayed by measuring NADPH oxidation at 340 nm and using glutathione oxidized (GSSG) as substrate [60].

Glutathione-S transferase (GST) activity was monitored spectrophotometrically at 340 nm by the formation of glutathione-CDNB-conjugate [61].

Glutathione (GSH) was measured by the procedure [62], partially modified [63], and adapted to the microtiter plate [64]. In the samples, both tGSH (total glutathione) and GSSG (oxidized glutathione) were also measured. For GSSG determination, the samples were derivatized by 2-vinylpyridine. The reaction was initiated by quickly adding 40 µL GR per well. The increase in absorbance was monitored at 415 nm. Standards were prepared to contain 0–100 mM GSH for tGSH and 0–8 mM GSSG. GSH levels were calculated by subtracting GSSG values from tGSH. The oxidative stress index (OSI) was calculated as:OSI = (2 GSSG/tGSH) × 100 

Trolox Equivalent Antioxidant Capacity (TEAC) was assayed by the neutralization made by the extract ABTS^+^ at 595 nm [65]. TEAC values were expressed as µM equivalent of Trolox (analogous to vitamin E).

Thiobarbituric Acid Reactive Substances (TBARS) were assayed considering that the samples utilized in the assay have malondialdehyde (MDA) that reacts with thiobarbituric acid (TBA). The reading was made in a microplate at 535 nm, using MDA as standard [66].

### 2.6. Treatment of Data

Six juveniles per sample were euthanized by lethal MS-222 immersion (100 mg L^−1^), weighted (Sartorius, MC210P, Germany), and photographed for curved length measurements [67] using NIS Elements software (Nikon). The following indices were calculated:-Survival (S, %): (final number of fishes/initial number of fishes) × 100, accounting for sampled juveniles-Specific growth rate (SGR, % day^−1^): Ln w_f_–Ln w_i_/t × 100, where w_f_ and w_i_ are the final and initial mean weight, and t is the experimental time in days.-Fulton’s Factor Condition Index: K = W/L^3^ × 10, where W and L are mean weight and length, respectively.


### 2.7. Statistical Analysis

The data were tested for normality and homoscedasticity using Shapiro–Wilk’s and Levene’s tests, respectively. When those conditions were not met, a Rank transformation was applied [68]. Mean comparisons for salinity levels on survival and growth indices were analyzed by t-test. Biochemical indices were compared using factorial ANOVA, considering pH and age (DAR) as fixed factors. Significant differences in ANOVA were assessed by the Newman-Keuls test. The minimum significance level was set at 5% (*p* < 0.05). Data were expressed as mean ± standard deviation. The statistical analyses were performed in a Statistica 7.0 software package.

Principal Component Analyses (PCA) were performed in R v.3.6.1 [69] to summarize and graphically visualize the results achieved. For this purpose, factoMineR v2.3 [70], factoextra v1.0.7 [71] and corrplot v0.8.4 [72] packages in R were used. Data values were standardized (mean = 0; sd = 1).

## 3. Results

### 3.1. Trial 1: Effect of pH and Salinity on Juveniles

Survival at 21 DAR was high (86.9–98.9%) in both salinity conditions (SW and BW) (Table 3). The highest survival was achieved at pH 6.5 in SW. Growth did not differ between pH levels in SW. However, final length, final weight, and SGR in BW were significantly higher at pH 8.0 (Table 3).

#### 3.1.1. Biochemical Oxidative Stress Indices

Enzymatic activities and biochemical oxidative stress indices are provided in Table 4 and Table 5 for seahorses maintained in SW, and Table 6 and Table 7 for those reared in BW. Differences caused by pH were only obtained for GSH (SW and BW) and OSI (BW).

Seawater—SW

SOD activities in seahorses exposed to pH 6.5 and 8.0 were statistically similar and remained rather constant thorough the whole experimental period, whereas DTD levels increased with age from 2 DAR (Table 4).

The activity of CAT decreased in early-developing juveniles (2 DAR), but increased afterwards, achieving the highest level at 21 DAR. However, the increase in CAT activity at pH 6.5 occurred earlier than at pH 8.0. G6PDH activity increased slightly with age, reaching the highest values in 21 DAR juveniles (Table 4).

The activity of GPx was not affected by pH level, but it declined significantly from 0 DAR to 7 DAR. Subsequently, GPx levels increased until the end of the experiment. GR and GST showed a similar pattern, progressively increasing during the experimental period as the seahorses grew older (Table 5).

Regarding glutathione metabolism, the pattern followed by GSH was similar to that for GR and GST. Nevertheless, the increase in GSH occurred earlier, at pH 6.5. On the other hand, GSSG levels increased from 2 DAR onwards. The relationship between glutathione forms (OSI) remained almost constant with development (Table 5).

TEAC values were similar across ages at both pH levels (Table 5). Finally, TBARS was not detected in newborn seahorses (0 DAR), nor in 2 DAR at pH 8.0; its value increased with age (Table 5), but it was not significantly affected by pH.

Brackish water—BW

The activity of SOD was affected by pH level (pH 6.5 > pH 8.0), but not by age. DTD values increased from 0 to 2 DAR, but remained stable afterwards (Table 6).

The activity of CAT increased with age up to 7 DAR, but the raise observed at pH 6.5 occurred earlier than that at pH 8.0 (7 and 14 DAR, respectively). Excluding 2 DAR juveniles (i.e., only one sample available), the levels in G6PDH were similar across pH levels and age (Table 6).

The activity of GPx was higher at 0 DAR and decreased significantly until 7 DAR, but increased afterwards. The GR activities in BW performed similarly to those in SW. There was an increase from 0 to 2 DAR, and activities remained similar afterwards. GST showed differences in neither pH nor age (Table 7).

GSH levels were significantly affected by both age and pH levels. GSH values at 14 and 21 DAR were significantly lower in the newborns kept at pH 6.5 than those kept at pH 8.0. GSSG levels increased according to age, reaching the highest values in 21 DAR juveniles. OSI was affected by both age and pH. The OSI values increased from 2 to 21 DAR for the newborns reared at pH 6.5, whereas OSI values were rather stable in the juveniles reared at pH 8.0 (Table 7).

TEAC was not affected by pH level, but decreased with age, being negligible in 21 DAR juveniles. TBARS was not affected by age or pH level.

#### 3.1.2. Global Assessment: SW and BW

The Principal Component Analysis (PCA) performed on SW and BW samples showed significant discrimination between both salinity groups on factors 1–2 representation, which explained 45.4 and 19.5% of total variability, respectively (Figure 1a). The main overall differences between both salinity groups corresponded to OSI, SOD, and DTD (positively associated with BW samples), and TEAC and GSH (positively associated with SW samples) values. Globally, TBARS and CAT levels were positively associated with age (length and weight).

The main differences across age and salinity groups (Figure 1b) were due to development, TBARS (especially in SW), CAT, and, to a lesser extent, GST. TEAC values were positively associated with early developmental stages, especially in BW groups.

The PCA representation for salinity–pH combinations (Figure 1c) showed smaller differences (smaller centroid distances) with pH in SW treatment compared to BW samples (see also Figure 1a). At low pH levels, OSI and SOD values were higher in BW samples, whereas SW newborns showed higher TEAC and survivals, and lower TBARS values.

### 3.2. Trial 2: Effect of Acidification on Juveniles

The final length and weight (49 DAR) of 21 DAR juveniles grown at pH 6.5 were 12% and 29% higher, respectively, than for juveniles maintained at pH 8.0 (Figure 2). Juveniles reared at pH 6.5 showed higher SGR than those reared at pH 8.0. However, survival (82.4 ± 10.6 for pH 6.5 and 72.5 ± 7.4 for pH 8.0) and condition factor (0.16 ± 0 for pH 6.5 and 0.17 ± 0.02 for pH 8.0) were similar in both treatments.

## 4. Discussion

The knowledge of fish responses to stressors during their lifespan is highly relevant for the understanding of potential physiological alterations in animals inhabiting natural habitats, as well as for the management of ex situ rearing [73,74]. In the present study, we assessed the impact of acidic conditions on the overall fitness of early developing seahorses *H. reidi* reared in seawater (SW) or brackish water (BW), as well as in more developed juveniles kept in SW. Our global results indicate that survival and growth were hampered in individuals kept in BW under acidic conditions. Compared to BW, acidic conditions in seawater (SW) led to higher survival and growth in both newborns and older juveniles. This is a highly interesting finding regarding its applicability to rearing systems and to studies carried out in changing environments.

### 4.1. Combined Effects of Salinity and pH on Early-Developing Juveniles

Seahorses born from the same breeding group show similar conditions at the time of the male’s pouch release [75]. In captivity, newborn batch size is generally 300–600 individuals [44,75]. Since our experimental needs were much higher, the experimental design included two sub-trials, one for BW and another for SW. Consequently, comparisons between salinities were not performed using univariate statistics.

Acidic pH conditions were strongly associated with changes in juvenile fitness. At pH 6.5, survival was enhanced in SW compared to BW. It is likely that newborn seahorses released at 33 salinity (at which the breeders were kept) had to deal with -challenges to satisfactorily perform the ion regulation process in BW compared to those in SW. Even though pregnant seahorses open their brooding pouches before newborn release in order to progressively adapt the newborns to external salinity conditions [76], the occurrence of ionocyte cells in *H. reidi* juveniles has not been observed before 6 DAR [51]. It has been reported in medaka (*Oryzias dancena*) that the complete acclimation of ionocytes to salinity changes requires approximately two weeks [77]. Cobia *Rachycentrum canudom* [78] and white seabass *Atractoscion nobilis* [79] deal with acidic exposure or ocean acidification by increasing the number of ionocyte cells and NKA activity, demonstrating an acid–base and ionic regulation interaction [80]. Thus, the capacity to deal with acidic conditions is linked with physiological effects of BW and acidic pH changes, as reported in older juveniles of the same species which were acutely exposed to these conditions [42].

The compensation of acidosis in marine fishes is principally adjusted by differential regulation of HCO_3_^−^ and H^+^– effluxes, which seems to be coupled to the influx of Na^+^ and Cl^−^ [37,81]. Thus, the acidic pH is a threat which is likely intensified by the low concentration of Na^+^ and Cl^−^ in BW conditions. This feature is also supported by the results achieved in seabass (*Dicentrarchus labrax*) acclimatized to ocean acidification in brackish water after dealing with ammonia challenges [37]. Since the ion regulation and acid–base equilibrium are intrinsic mechanisms, these two alterations together triggered losses in the fitness of *H. reidi* juveniles.

### 4.2. Biochemical Indices: Seawater–SW (S33) and Brackish Water–BW (S11)

The oxidative status in *H. reidi* juveniles was more affected by the growth/age of seahorses than by pH level. For example, G6PDH and GR showed an age-dependent increase. GSH is restored from GSSG by GR, which uses NADPH as a donator for H^+^. The NADPH is produced by G6PDH activity; thus, the G6PDH function is essential in GR activity [82]. In fishes, an age-dependent relationship has been reported between GPx and GR activity to maintain intracellular GSH homeostasis [83,84], which is in accordance with our results. In newborn seahorses, DTD displayed the same age-dependent trend as CAT, GPx, GR, and GST, independently of salinity level or acidification conditions. DTD activity is related to the reduction of quinones to hydroquinones through a transfer of two electrons of reduced cofactors, NADH or NADPH [85]. DTD activity likely avoids the potential increase of free radicals with growth, whereas other enzymes (e.g., CAT, GPx, and GST) act to scavenge undesirable ROS levels. All age-dependent responses referred to above might be linked to food composition, as well as to fish growth.

The oxidative balance may change with fish growth or by environmental conditions [86]. However, the antioxidant compounds received by the fish from the ingested prey may change with dietary changes [36,44,87,88], influencing the oxidative status of juveniles. Moreover, it is known that alterations in the oxidative status are important to determine ontogeny events, as well as for the cell signaling involved in the development of tissues and physiological systems [89,90,91]. Thereby, age, feeding, and ontogenic changes are pivotal factors in the initial development of fishes, very likely supporting the results achieved in the present study on the oxidative status in seahorse juveniles.

Regarding growth and oxidative status, it is important to highlight that fishes facing environmental challenges decrease their aerobic energy generation [33,34,92,93] in order to reduce the production of free radicals [94]. These findings and our results are consistent with a potential scenario of metabolism reduction in seahorse juveniles kept in BW with pH 6.5. Furthermore, the resilient oxidative status revealed in SW at pH 6.5 led to slight improvements in survival and growth in older juveniles (Trial 2). Salinity changes trigger alterations to the immune system, which is relevant for the oxidative status, survival, and growth of fishes [15,95,96,97].

Both SOD and CAT are antioxidant enzymes involved in the first line of defense against ROS. The enzyme SOD reduces the anion superoxide (O2^•−^) to H_2_O_2_, which is further transformed by CAT into harmless molecules (H_2_O and O_2_). The stable SOD activity in juveniles from trial 1, independently of the salinity level, suggests that acidification did not increase the production of O2^•−^ in cellular metabolism. Stable SOD activities have also been reported in flounder (*Paralichthys olivaceus*) larvae [98] and oysters (*Crassostrea gigas* and *C. angulata*) exposed to an acidic environment [99]. Additionally, the interaction between H^+^ and O_2_^•−^ was also a possible way for SOD inactivation to occur [38].

CAT activities showed an age-dependent pattern, increasing with growth and suggesting that ontogeny is the main factor driving this activity in *H. reidi* juveniles. This statement agrees with the results achieved regarding the first developmental stages of sturgeon (*Acipenser naccarii)* [100]. In the same way, Robergs [101,102] pointed out that many interactions can occur with H^+^ within a pH range from 6.0 to 7.0. Hence, we hypothesize that H^+^ also enhanced NADPH production and the stability in SOD and CAT activities, which, very likely, was the consequence of a direct interaction between H^+^ and O2^•−^ [38]. On the other hand, intense alterations in H^+^ and other ions can lead to mitochondrial hyperpolarization and cell death [103]. We suggest that the same feature occurred in seahorses kept in BW at pH 6.5.

Acidification did not affect GPx activity in trial 1. However, Cui et al. [98] registered an increase in this activity in flounder larvae (*Paralichthys olivaceus*) due to acidification. However, we observed that glutathione (GSH) increased more quickly in SW, whereas GSH, GSSG, and OSI increased more quickly in BW.

GSH and enzymes related to its metabolism (GPx, GR, and GST) have an important role against ROS increase [36,40]. We observed that GSH content in BW conditions was lower at pH 6.5 than at pH 8.0, but it did not change significantly with the juveniles’ age. However, it has been reported in other species, such as sturgeon (*A. naccarii*), that GSH increased with growth during the free embryo stage and drop-stabilized once juveniles began exogenous feeding (i.e., 21 days after fertilization) [100]. It is important to note that seahorses are fish undergoing a large and protected embryogenesis, in which the embryos are nourished by males until being released from the brood pouch to the external environment [104,105]. Newly released juveniles almost completely lack a yolk sac and show exogenous feeding. These features might explain the stability of GSH levels in growing juveniles.

The Oxidative Stress Index (OSI) relates the proportion of GSSG to the total amount of GSH (tGSH). Our study revealed a status of low stress in seahorses reared in SW, independently of water acidification, and a stressful condition in seahorses kept in BW at pH 6.5. On the other hand, OSI levels represent an inverse relation with total antioxidant capacity (TEAC). As mitochondrial respiration (i.e., ROS production) increased with growth, TEAC levels dropped dramatically in seahorses kept in BW, and the oxidation degree (OSI) reached its highest levels by the end of the experimental period (Trial 1).

In our study, glutathione levels improved with age in juveniles kept in SW pH 6.5. Hence, it is feasible that both juveniles and adults kept in acidic environments have a better oxidative status (i.e., fitness) than individuals kept in a natural environment (pH 8.0).

We did not observe differences in TBARS levels regarding treatment conditions. However, considering the overall alterations (including growth and, to a lesser extent, survival), it is feasible that the juveniles kept in BW pH 6.5 underwent a metabolic depression. In this regard, low LPO values were reported in fish subjected to stress [106,107].

### 4.3. Global Assessment (PCA)

The multivariate approach revealed a global view of the acidic effects on juveniles submitted to both SW and BW conditions. In juveniles from trial 1, the development of juveniles under BW conditions showed low survival and growth compared to SW. This finding disagrees with the optimal salinity for growth (10–20 salinity) previously assessed in the same species [29,39]. These discrepancies in growth might rely on differences in newborn quality, as revealed by some biochemical indices, in 0 DAR juveniles (e.g., TBARS = 0 ± 0 and 1.56 ± 1.0 nmol MDA mg protein^−1^ for SW and BW, respectively). However, previous studies have reported small inter-batch differences in the performance of *H. reidi* newborns produced at our facilities [75]. Moreover, the survival rates attained under the worse environmental scenario in trial 1 (pH 6.5 in brackish water) were similar to those reported in cultures performed at 10–20 salinity [29].

PCA revealed that juveniles kept in BW were more susceptible to SOD increases, reinforcing the hypothesis of lower resilience of fish grown under acidic conditions at low salinity. Similarly, SOD activities increased in sand smelt (*Atherina presbyter*) larvae kept in slightly acidified media, but dropped under more drastic acidification [108]. Since SOD, OSI, and DTD were associated with BW in the PCA, the interactions of these parameters could be responsible for the mortalities and lower growth observed in BW at pH 6.5.

TEAC levels were positively correlated with survival in SW, whereas OSI was negatively correlated with survival in BW. Increases in GSH could help to reduce the harmful effect of ROS and intracellular homeostasis maintenance in stressing conditions such as acidic environments [109]. However, the levels of GSH in BW were almost two-fold higher in seahorses reared at pH 8.0 compared to those reared at pH 6.5. Hence, even in an estuarine species such as *H. reidi*, acidification conditions accompanied by reduced salinities would hardly be tolerated. This statement was already recognized in *D. labrax* [37].

Regarding oxidative stress, TBARS levels were correlated with age/growth. Similar results were reported in *D. labrax* larvae by Maulvault et al. [107], who observed increases in malondialdehyde (MDA) in warm and acidification conditions, accompanied by a growth improvement. However, juveniles in SW displayed lower TBARS levels than those in BW and acidic conditions. It is likely that the higher survivals in SW are enhanced by higher TEAC and GSH levels.

### 4.4. Trial 2: Effect of pH on the Growth of Seahorse Juveniles

Since age and size are intrinsic factors to consider for the understanding of environmental stress in fishes [110,111], we carried out a second experiment in older juveniles. Given that trial 1 revealed lower resilience to acidic conditions in juveniles in BW, the second trial was carried out only in a SW environment.

Explanations supporting growth improvement in juveniles kept in SW at pH 6.5 should consider parasitology and energy generation. Regarding the former, the effect of acidic conditions on ectoparasites is well-documented. Ectoparasite growth is impaired under acidic conditions. Consequently, parasitology threats would be reduced in seahorses reared at low pH [4,112].

Regarding cell energy, pH levels may alter the matrix membrane potential in mitochondria, which may enhance ATP-synthase function [113] and generate more energy (ATP). It is important to highlight that G6PDH activity is linked to growth factors [114]. Although enzymatic analyses were not performed in this trial, the increasing G6PDH activity with age (Trial 1) suggests that the growth would be enhanced in older juveniles reared under acidic conditions (Trial 2). This assumption agrees with the increase in growth induced at pH 6.3 in turbot *Psetta maxima* when compared to upper pH levels tested (up to pH 8.8) [115]. The good results obtained in the oxidative status and growth of juveniles raised in SW at pH 6.5 are related to acidic chemistry, since H^+^ is involved in aerobic and non-aerobic energy metabolism, affecting mitochondrial proton leak while interacting with metabolites of non-mitochondrial energy generation [101,102,116]. This can trigger benefits for cell energy and oxidative status.

## 5. Conclusions

The development of *Hippocampus reidi* juveniles grown at low salinity (BW) caused SOD, DTD, OSI increases, and TEAC consumption, revealing a stressful condition. Additionally, survival and growth were hampered in juveniles grown at pH 6.5. Consequently, our results indicate that pH oscillations should be avoided in rearing performed in BW, since seahorses showed low resilience to acidic conditions. We suggest that *H. reidi* juveniles be reared in SW under acidic conditions (pH 6.5), which increased survival and growth. Thus, the results of the present study can be applied in order to enhance the early rearing of *H. reidi* in captivity.

## Figures and Tables

**Figure 1 animals-12-03227-f001:**
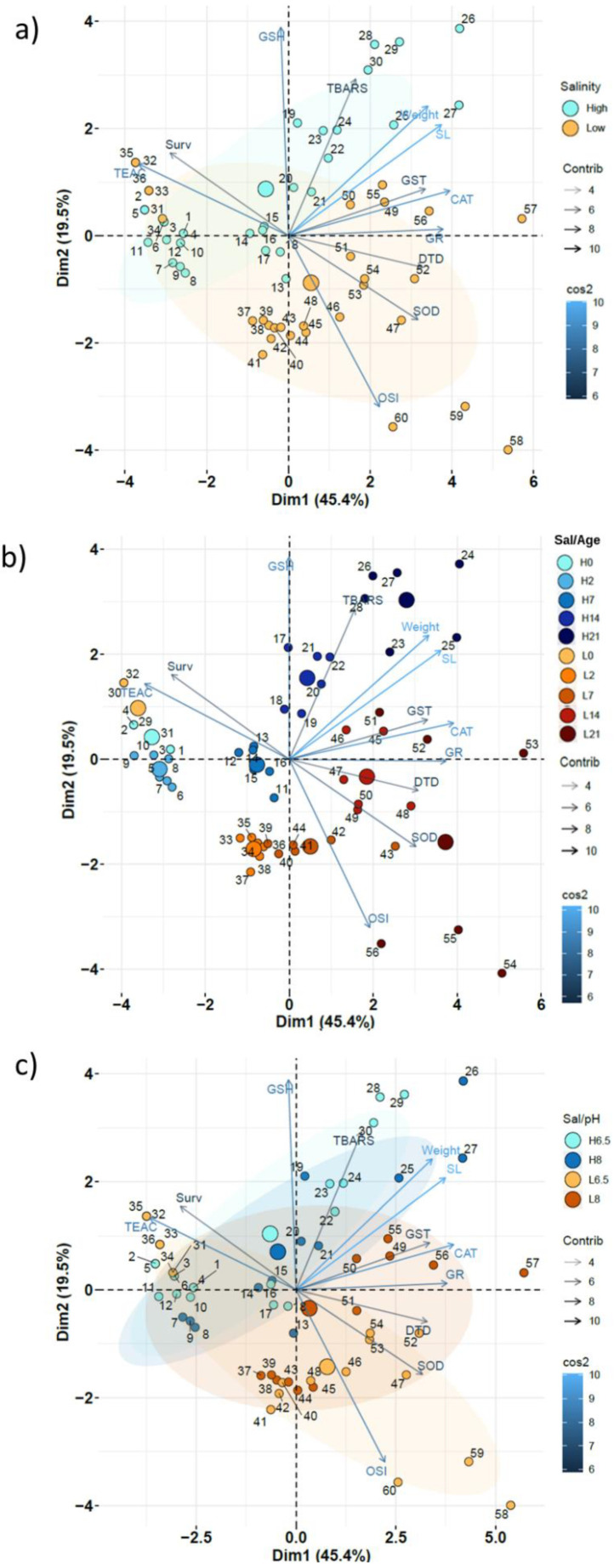
Trial 1—PCA plots of survival, enzymatic activities, and enzymatic indices in seahorse juveniles *H. reidi.* Sample IDs and the variables with the highest contributions (cos^2^ > 0.5) are indicated. Ellipses correspond to centroid values ± 1 s.d. (shaded areas). (**a**) Seawater (SW = High—33) vs. brackish water (BW = Low—11) groups. (**b**) All salinity levels (33 and 11) and ages (0–21 DAR). (**c**): All salinity (SW—33 and BW—11) and pH (6.5 and 8) levels.

**Figure 2 animals-12-03227-f002:**
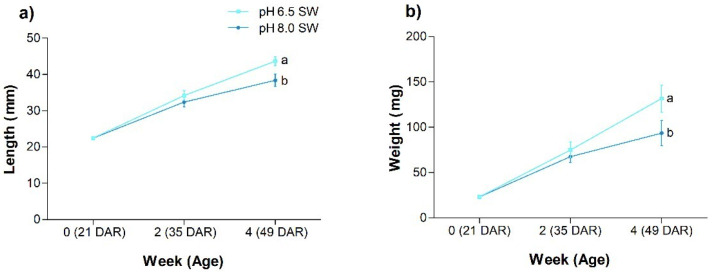
Trial 2—growth, (**a**) length and (**b**) weight (mean ± standard deviation) in 21 DAR seahorse juveniles *H. reidi* maintained for four weeks in SW at pH 6.5 and 8. Different letters indicate significant differences between pH levels at each age.

**Table 1 animals-12-03227-t001:** Trial 1—water conditions (mean ± standard deviation) for the maintenance of *H. reidi* (age 0–21 DAR—days after the male’s pouch release) reared in seawater (SW—salinity 33) or brackish water (BW—salinity 11) at pH 6.5 or 8.0 for 21 days.

	SW		BW	
	pH 6.5	pH 8.0	pH 6.5	pH 8.0
Salinity (‰)	33 ± 1		11 ± 1	
pH	6.6 ± 0.01	8.1 ± 0.0	6.5 ± 0.01	7.8 ± 0.01
Alkalinity (mg CaCO_3_ L^−1^)	22 ± 5 ^b^	138 ± 5 ^a^	20 ± 5 ^b^	62 ± 5 ^a^
Temperature ( °C)	26.1 ± 0.1		26.1 ± 0.2	
Oxygen (mg O_2_ L^−1^)	6.53 ± 0.03		6.56 ± 0.01	
TAN (mg N- NH_4_ ^+^ +NH_3_ L^−1^)	0.21 ± 0.04		0.13 ± 0.02	
Nitrite (mg N-NO_2_ L^−1^)	0.02 ± 0.0		0.05 ± 0.05	
Nitrate (mg N-NO_3_ L^−1^)	0.13 ± 0.03		0.11 ± 0.01	

Within salinities, different letters indicate significant difference (*p* < 0.05) (Student’s t test).

**Table 2 animals-12-03227-t002:** Trial 2—Water conditions (mean ± standard deviation) for the maintenance of *H. reidi* (21–49 DAR—days after the male’s pouch release) reared in seawater (SW—salinity 33) at pH 6.5 or 8.0 for four weeks.

	pH 6.5	pH 8
Salinity (‰)	33 ± 1	
pH	6.6 ± 0.01	8.1 ± 0.0
Alkalinity (mg CaCO_3_ L^−1^)	21 ± 4 ^b^	142 ± 7 ^a^
Temperature (°C)	26.0 ± 0.5	
Oxygen (mg O_2_ L^−1^)	6.5 ± 0.5	
TAN (mg N-NH_4_ ^+^ +NH_3_ L^−1^)	0.3 ± 0.1	
Nitrite (mg N-NO_2_ L^−1^)	0.1 ± 0.05	
Nitrate (mg N-NO_3_ L^−1^)	0.2 ± 0.05	

Different letters indicate significant differences (*p* < 0.05) (Student’s t test).

**Table 3 animals-12-03227-t003:** Trial 1—Effects of pH and salinity on survival, growth, specific growth rate (SGR) and condition factor (K) (mean ± standard deviation) in *H. reidi* juveniles reared in seawater (SW—salinity 33) or brackish water (BW—salinity 11) at pH 6.5 or 8.0 for 21 days (age 0–21 DAR—days after the male’s pouch release).

	SW		BW	
	pH 6.5	pH 8.0	pH 6.5	pH 8.0
Survival (%)	98.9 ± 0.48 ^a^	96.9 ± 0.96 ^b^	86.9 ± 2.2 ^b^	92.2 ± 2.2 ^a^
Final length (mm)	27.5 ± 0.86	26.9 ± 1.10	19.6 ± 0.2 ^b^	22.8 ± 0.7 ^a^
Final weight (mg)	37.8 ± 2.7	35.8 ± 3.5	17.6 ± 0.4 ^b^	27.9 ± 2.2 ^a^
SGR (%)	5.1 ± 0.2	5.0 ± 0.1	4.0 ± 0.0 ^b^	4.8 ± 0.1 ^a^
K	0.18 ± 0.01	0.18 ± 0.01	0.23 ± 0.03	0.23 ± 0.00

Within salinities, different letters indicate significant differences (*p* < 0.05) (Student’s t test).

**Table 4 animals-12-03227-t004:** Trial 1 (SW)—Enzymatic activities (mean ± standard deviation) in *H. reidi* juveniles reared in seawater (SW—salinity 33) at pH 6.5 or 8.0 for 21 days (age 0–21 DAR—days after the male’s pouch release).

		Age (Days after Male’s Pouch Release—DAR)					ANOVA (*p*)		
	pH	0	2	7	14	21	age	pH	Age × pH
SOD	6.5	51.5 ± 10.2	51.6 ± 19.9	45.7 ± 1.8	49.9 ± 3.2	47.4 ± 2.5	0.159	0.613	0.478
	8.0		47.0 ± 0.7	58.1 ± 16.4	52.2 ± 2.8	75.1 ± 11.0			
DTD	6.5	3.1 ± 1.0 ^ab^	2.7 ± 1.0 ^b^	4.2 ± 0.5 ^ab^	4.2 ± 0.2 ^ab^	4.64 ± 0.3 ^a^	0.001	0.788	0.238
	8.0		3.4 ± 0.2 ^ab^	4.3 ± 0.5 ^ab^	3.1 ± 0.1 ^ab^	4.7 ± 0.5 ^a^			
CAT	6.5	3.6 ± 0.6 ^cd^	2.6 ± 0.7 ^d^	4.5 ± 1.4 ^c^	7.7 ± 1.2 ^ab^	11.4 ± 0.5 ^a^	<0.001	0.252	0.438
	8.0		2.5 ± 0.0 ^d^	3.6 ± 0.3 ^cd^	5.0 ± 1.1 ^bc^	12.1 ± 1.5 ^a^			
G6PDH	6.5	1.0 ± 0 ^b^	1.22 ± 0.3 ^b^	SNA	2.0 ± 0.0 ^a^	2.7 ± 0.3 ^a^	<0.001	-	0.958
	8.0		1.1 ± 0.1 ^b^	1.3 ± 0.1 ^b^	SNA	2.6 ± 0.0 ^a^			

Within enzymatic activities, different letters indicate significant differences (two-way ANOVA; Newman-Keuls test). SNA: sample not analyzed.

**Table 5 animals-12-03227-t005:** Trial 1 (SW)—Glutathione metabolism (mean ± standard deviation) in *H. reidi* juveniles reared in seawater (SW—salinity 33) at pH 6.5 or 8.0 for 21 days (age 0–21 DAR—days after the male’s pouch release).

		Age (Days after Male’s Pouch Release—DAR)					ANOVA (*p*)		
	pH	0	2	7	14	21	Age	pH	Age × pH
GPx	6.5	420 ± 240 ^ab^	398 ± 45 ^a^	168 ± 54 ^c^	227 ± 22 ^abc^	351 ± 36 ^ab^	<0.001	0.231	0.934
	8.0		344 ± 53 ^ab^	129 ± 52 ^c^	193 ± 10 ^bc^	342 ± 23 ^ab^			
GR	6.5	5.5 ± 0.3 ^c^	5.2 ± 0.8 ^c^	7.2 ± 0.3 ^bc^	8.6 ± 0.7 ^ab^	10.4 ± 0.8 ^a^	<0.001	0.640	0.901
	8.0		5.2 ± 0.2 ^c^	7.5 ± 0.8 ^bc^	7.9 ± 0.9 ^b^	10.0 ± 1.4 ^a^			
GST	6.5	5.0 ± 1.2 ^d^	6.3 ± 0.1 ^d^	10.6 ± 1.3 ^cd^	18.1 ± 3.5 ^b^	17.3 ± 2.3 ^bc^	<0.001	0.288	0.087
	8.0		5.7 ± 0.1 ^d^	12.1 ± 4.1 ^cd^	16.0 ± 0.0 ^bc^	24.3 ± 3.6 ^a^			
GSH	6.5	59.1 ± 2.7 ^ab^	48.4 ± 0.5 ^b^	57.6 ± 4.4 ^ab^	68.5 ± 11.2 ^a^	73.9 ± 3.5 ^a^	<0.001	0.031	0.303
	8.0		45.7 ± 0.9 ^b^	56.2 ± 3.4 ^ab^	58.1 ± 2.6 ^ab^	60.8 ± 9.6 ^ab^			
GSSG	6.5	2.6 ± 0.1 ^ab^	2.3 ± 0.2 ^b^	3.5 ± 0.3 ^ab^	3.4 ± 0.1 ^ab^	3.1 ± 0.4 ^ab^	0. 006	0.379	0.175
	8.0		2.7 ± 0.0 ^ab^	2.9 ± 0.7 ^ab^	3.6 ± 0.8 ^ab^	3.9 ± 0.2 ^a^			
OSI	6.5	8.7 ± 0.7	9.7 ± 0.8	12.3 ± 1.9	10.0 ± 1.4	8.4 ± 1.4	0.379	0.095	0.171
	8.0		11.9 ± 0.3	10.6 ± 3.1	12.5 ± 3.3	12.9 ± 2.7			
TEAC	6.5	207 ± 27 ^a^	183 ± 0.7 ^ab^	76 ± 43 ^b^	73 ± 38 ^b^	99 ± 25 ^b^	<0.001	0.714	0.543
	8.0		173 ± 32 ^ab^	88 ± 52 ^b^	121 ± 47 ^ab^	76 ± 24 ^b^			
TBARS	6.5	0.00 ± 0.00 ^c^	0.02 ± 0.02 ^bc^	0.06 ± 0.05 ^bc^	0.18 ± 0.11 ^ab^	0.14 ± 0.05 ^ab^	<0.001	0.918	0.540
	8.0		0.0 ± 0.0 ^c^	0.06 ± 0.03 ^bc^	0.17 ± 0.09 ^ab^	0.34 ± 0.2 ^a^			

Within enzymatic activities, different letters indicate significant differences (two-way ANOVA; Newman-Keuls test).

**Table 6 animals-12-03227-t006:** Trial 1 (BW)—Enzymatic activities (mean ± s.d.) in *H. reidi* juveniles reared in brackish water (BW—salinity 11) at pH 6.5 or 8.0 for 21 days (age 0–21 DAR—days after the male’s pouch release).

		Age (Days after Male’s Pouch Release—DAR)					ANOVA (*p*)		
	pH	0	2	7	14	21	Age	pH	Age × pH
SOD	6.5	39.6 ± 17.6 ^b^	68.5 ± 4.6 ^a^	100.7 ± 32.4 ^a^	91.3 ± 14.6 ^a^	89.3 ± 33.3 ^a^	0.012	0.311	0.331
	8.0		68.9 ± 6.8 ^a^	62.9 ± 14.4 ^a^	71.9 ± 4.5 ^a^	101.6 ± 25.8 ^a^			
DTD	6.5	1.9 ± 2.1 ^b^	4.1 ± 0.6 ^a^	4.5 ± 0.3 ^a^	4.44 ± 0.6 ^a^	4.3 ± 0.1 ^a^	0.019	0.245	0.775
	8.0		4.3 ± 0.9 ^a^	4.5 ± 0.1 ^a^	5.1 ± 0.5 ^a^	4.6 ± 0.6 ^a^			
CAT	6.5	4.7 ± 0.2 ^b^	5.5 ± 1.2 ^b^	8.2 ± 2.3 ^ab^	11.9 ± 1.0 ^a^	10.4 ± 4.7 ^ab^	<0.001	0.288	0.374
	8.0		5.0 ± 0.4 ^b^	5.3 ± 0.5 ^b^	8.8 ± 1.8 ^ab^	13.7 ± 4.1 ^a^			
G6PDH	6.5	2.0 ± 0.4	SNA	1.8 ± 0.3	1.8 ± 0.5	1.6 ± 0.6	0.147	-	0.110
	8.0		1.3 ± 0.1	1.0 ± 0.3	2.2 ± 0.5	2.2 ± 0.4			

Different letters indicate significant differences (two-way ANOVA; Newman-Keuls test). SNA: sample not analyzed.

**Table 7 animals-12-03227-t007:** Trial 1 (BW)—Glutathione metabolism (mean ± standard deviation) in *Hippocampus reidi* juveniles reared in brackish water (BW—salinity 11) at pH 6.5 or 8.0 for 21 days (age 0–21 DAR—days after the male’s pouch release).

		Age (Days after the Male’s Pouch Release—DAR)					ANOVA (*p*)		
	pH	0	2	7	14	21	Age	pH	Age × pH
GPx	6.5	486 ± 110 ^a^	271 ± 41 ^ab^	177 ± 2 ^b^	313 ± 21 ^ab^	226 ± 72 ^b^	<0.001	0.442	0.485
	8.0		256 ± 39 ^b^	171 ± 31 ^b^	323 ± 128 ^ab^	354 ± 59 ^ab^			
GR	6.5	7.4 ± 1.8 ^b^	7.4 ± 0.9 ^b^	10.9 ± 2.8 ^ab^	12.6 ± 0.9 ^a^	11.3 ± 3.3 ^a^	0.008	0.550	0.462
	8.0		7.1 ± 0.3 ^b^	8.3 ± 0.6 ^ab^	10.7 ± 1.9 ^a^	13.4 ± 3.9 ^a^			
GST	6.5	13.1 ± 4.3	14.9 ± 4.5	21.0 ± 6.1	21.6 ± 8.2	14.4 ± 3.2	0.096	0.777	0.171
	8.0		15.3 ± 2.1	12.7 ± 2.7	23.2 ± 6.0	19.7 ± 5.7			
GSH	6.5	48 ± 5.5 ^ab^	27.3 ± 0.1 ^c^	37.2 ± 5.7 ^c^	32.9 ± 4.3 ^c^	24.0 ± 0.7 ^c^	<0.001	<0.001	<0.001
	8.0		31.5 ± 3.3 ^c^	32.5 ± 3.8 ^c^	56.5 ± 7.8 ^a^	56.7 ± 0.3 ^a^			
GSSG	6.5	1.1 ± 1.5 ^d^	2.5 ± 0.5 ^cd^	2.6 ± 0.01 ^cd^	2.0 ± 0.01 ^d^	4.3 ± 0.4 ^a^	<0.001	0.072	0.009
	8.0		2.5 ± 0.2 ^cd^	3.2 ± 0.6 ^abc^	2.8 ± 0.2 ^bc^	3.4 ± 0.3 ^ab^			
OSI	6.5	4.7 ± 6.7 ^e^	18.2 ± 3.8 ^bc^	15.6 ± 1.4 ^bc^	13.6 ± 2.0 ^bc^	36.0 ± 2.1 ^a^	<0.001	0.001	<0.001
	8.0		16.0 ± 0.6 ^bc^	19.4 ± 1.6 ^b^	10.1 ± 1.0 ^de^	11.5 ± 0.6 ^cde^			
TEAC	6.5	210 ± 15 ^a^	55 ± 53 ^abc^	40 ± 22 ^abcd^	48 ± 39 ^abcd^	2.2 ± 3.8 ^cd^	<0.001	0.529	0.748
	8.0		92 ± 0 ^ab^	67 ± 48 ^ab^	33 ± 23 ^bcd^	0 ± 0 ^d^			
TBARS	6.5	1.56 ± 1.0	1.21 ± 0.1	0.15 ± 0.2	0.63 ± 0.4	0.44 ± 0.22	0.081	0.164	0.300
	8.0		1.25 ± 0.4	1.26 ± 0.3	0.38 ± 0.3	1.23 ± 0.94			

Different letters indicate significant differences (two-way ANOVA; Newman-Keuls test). SNA: sample not analyzed.

## Data Availability

Data not listed in this collection are available from the authors upon reasonable request.

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
