# Peer review of "Implications of Salinity and Acidic Environments on Fitness and Oxidative Stress Parameters in Early Developing Seahorses Hippocampus reidi"

_animals, 2022, doi:10.3390/ani12223227_

Round 1

Reviewer 1 Report

The authors characterized the effects of salinity and acidic environments on fitness and oxidative stress parameters in early developing seahorses Hippocampus reidi. The seahorse juveniles were exposed to 4 experimental conditions using an experimental design combining the 2 water parameters, pH 6.5 and 8.0, salinity 11 and 33. They found that the concurrence of acidic pH (6.5) and low salinity has harmful effects on the fitness and development of seahorse juveniles. The manuscript is well written. However, it can’t be accepted in its current form.

Lines 25 – 29, these sentences are hard to understand. I suggest the authors formulate 4 symbols to represent the experimental groups and perform one-way ANOVA with multiple comparisons to analyze significant difference in the measurements for different experimental groups.

Table 3, the authors compared means of the indices including survival rate, final weight, final length and SGR between the two experimental groups of the same salinity level. However, to fully understand effects of the environmental parameters, the four experimental groups should be analyzed together.

For the tables, the meaning of the upper scripts should be explained in the footnotes.

Lines 135, 146 and 567, the Latin name should be italic.

Reviewer 2 Report

Dear authors

The manuscript of Carneiro et al about the implications of salinity and acidity in seahorses showed that the best conditions for seahorses (H. reidi) are reached when raised in seawater under low pH. The paper is well written and several parameters were determined. Although there are some research papers published in this area already, authors combined saltwater with brackishwater at different pH, except for the second trial. 

In some tables it is difficult to understand where the statistical letters stand for, meaning for example table 4: CAT is significantly different in age, pH and age vs pH. For which comparison are the letters?

Table 1 and 2: caption, information about statistical differences by letters is missing

Table 4,5,6 and 7: p values are not consistently in italics

table 7: GR statistical letters are in capital

Some minor revisions:

line 76, 98 and 106: please refer to the species with H. instead of Hippocampus. Check throughout the manuscript

Line 135: species H. reidi in italics

Line 146: species in italics

Line 159: please correct sump in pump

Line 231: remove the brackets from the ref

Line 289: suggestion: ... and tables 6 and 7...

Line 317: TBARS was also not detected in 2DAR, please add

Line 409: Please delete in and start the sentence with Cobia

Line 420, 423: species italics, please check further the manuscript (Lines 536, 567, 571)
